# Views and Perceptions of People Aged 55+ on the Vaccination Programme for Older Adults in the UK: A Qualitative Study

**DOI:** 10.3390/vaccines11040870

**Published:** 2023-04-20

**Authors:** Taru Silvonen, Jo Kesten, Christie Cabral, Jo Coast, Yoav Ben-Shlomo, Hannah Christensen

**Affiliations:** 1Population Health Sciences, Bristol Medical School, NIHR Health Protection Research Unit in Behavioural Science and Evaluation, University of Bristol, Oakfield House, Bristol BS8 2BN, UK; 2The National Institute for Health and Care Research Applied Research Collaboration West (NIHR ARC West), University Hospitals Bristol and Weston NHS Foundation Trust, Bristol BS1 2NT, UK; 3Population Health Sciences, Bristol Medical School, University of Bristol, Bristol BS8 2PS, UK; 4Health Economics Bristol, Population Health Sciences, Bristol Medical School, University of Bristol, Bristol BS8 1NU, UK

**Keywords:** vaccination, vaccine perceptions, vaccination motives, older adults, focus groups, qualitative research, influenza, pneumococcal, shingles, COVID-19

## Abstract

Vaccination reduces the risks related to infectious disease, especially among more vulnerable groups, such as older adults. The vaccines available to older adults in the UK through the government-funded programme currently include influenza, pneumococcal, shingles and COVID-19 vaccines. The purpose of the programme is disease prevention and improving wellbeing among the ageing population. Yet, the target population’s views of the programme remain unknown. This paper aims to increase the understanding of older adults’ perceptions of the vaccination programme available in the UK. A total of 13 online focus groups (56 informants) were carried out for this qualitative study. The findings indicate that getting vaccinated involves personal decision-making processes, which are influenced by previous experiences and interpersonal interactions. Factors related to the wider community and culture are less prominent in explaining vaccination decisions. However, opportunistic vaccination offers, a lack of information and a lack of opportunities to discuss vaccines, especially with healthcare professionals, are prominent factors. The study provides in-depth data about the rationale behind older adults’ vaccination decisions in the UK. We recommend that the provision of information and opportunities to discuss vaccines and infectious disease be improved to enable older adults’ to make better informed decisions regarding the vaccines available to them.

## 1. Introduction

Vaccinating older adults is a key public health intervention addressing the risks of serious outcomes or complications related to infectious diseases, such as influenza [1,2], pneumonia [3] and shingles [4]. However, vaccine uptake among older adults varies by country and by vaccine, often remaining low across certain age ranges in several countries [5,6] including the UK [4,7]. Factors that are linked to higher uptake among older people include existing health conditions [2], previous uptake of the same [8] or different routine vaccines [9], recommendations from healthcare professionals (HCPs) [8,10,11] and free vaccine offers [5], as well as shared normative beliefs that are in favour of vaccination in general [1].

Low uptake in general has been linked to access barriers, such as lack of transport [1], the cost of vaccination [12] and physicians’ concerns about vaccination [8]. A number of personal reasons for non-vaccination have been identified by previous studies: fear of side effects [3], disliking injections and vaccines being perceived as unnecessary or ineffective [2], concerns that vaccines might result in illness [1,13], general complacency or overconfidence about potential risks of a vaccine [12], and older adults’ low perceived risk of illness [8]. Furthermore, demographic and socio-economic factors have been linked to low vaccine uptake in the 74+ age group: women and those living in deprived areas or who are unmarried are less likely to get vaccinated [14]. Some ethnic groups have also been shown to be more vaccine-hesitant than others, among the general adult population [15] and among older adults in care facilities [16].

There is little evidence on the vaccination programme as a whole or the views of older adults aged 55 and over. Previous research focuses either on specific vaccines across adult populations [17] or one specific vaccine, such as influenza [2,8,18], within a more limited age range [14,19]. While there has been an increase in qualitative research on vaccine uptake in recent years, there are few qualitative studies that explore people’s views and perceptions beyond the COVID-19 vaccines or hesitant populations [20,21]. This qualitative research project addresses these knowledge gaps by focusing on the whole vaccination programme available to older adults through the National Health Service (NHS) in the UK. The aim of this research is to increase the understanding of older adults’ perceptions of the government-funded vaccination programme [22] for this age group, which currently consists of vaccines for influenza (offered annually to those aged 50+ years), shingles (available to 70–79 year olds), pneumococcal disease (available to those aged 65+ years) and COVID-19 (available to those aged 50+ years).

## 2. Materials and Methods

Thirteen online focus groups (FGs) were conducted across the UK. Participant recruitment was carried out between July 2021 and May 2022. Data collection was completed between October 2021 and May 2022. Ethical approval for this study was granted by the Faculty Research Ethics Committee (FREC) for the Faculty of Health Sciences at the University of Bristol (118051, 8529 latest amendment).

### 2.1. Methodological Approach

FGs were an appropriate method for this research because group discussions offer a setting for exploring a broad range of views and experiences, providing rich qualitative data [23]. FGs were chosen over other qualitative methods, such as one-to-one interviews, as they provide participants with an opportunity to converse with their peers. We felt that discussions among participants could draw out differences in what might affect people’s vaccination perceptions and behaviour. We chose to conduct the FGs online to minimize risks to both participants and researchers during ongoing and changing restrictions for face-to-face interactions due to COVID-19.

### 2.2. Participant Recruitment

To ensure participant recruitment from across the UK, we selected 15 research locations, one in each of the nine English regions and two in each of the devolved nations. We followed a three-step process for the areas in England: (i) identify five of the most urban regions, (ii) select the most ethnically diverse local authority area, (iii) identify the most ethnically diverse local authority area out of the more rural areas in each of the remaining four regions. A similar process was followed for two areas in each of the devolved nations, selecting the most ethnically diverse area in the most urban region and the most ethnically diverse area amongst the rural regions. This made it possible to explore differences across regions, as well as urban and less urban areas, while ensuring we were recruiting in ethnically diverse areas. The most ethnically diverse areas tend to be more urban, which is why the areas selected in the rural regions (either classified as urban with significant rural areas, largely rural, town and country living, or countryside living) are referred to as less urban throughout this article. Previous research shows that living in urban or rural areas can affect access to services [14,24], and vaccine uptake and confidence varies by ethnicity [15,16,25]. The research locations are listed in Table 1 below.

A call for participants (CfP) was shared through community-based organisations, such as faith groups, charity organisations and social groups focused on older adults in each of the 15 research locations. In the later stages, Facebook advertising and sharing the CfP through Twitter and print media supported participant recruitment. A CfP tailored to less vaccine-confident individuals was shared through four print newspapers, as well as Facebook advertising. This resulted in a few new expressions of interest, but no new consented participants. Individuals were guided to the project webpage and online pre-screening questionnaire. The pre-screening questionnaire asked potential participants to share their age and first part of postcode to determine eligibility; only those aged 55+ years and in our pre-specified research locations were eligible. When this research was designed, the government-funded vaccine offer for adults started at 65 years. However, due to new vaccine products being aimed at a slightly younger age bracket, we included those aged 55+ in this research. The questionnaire also asked for information that has previously been found to affect vaccination uptake, so that in the event of multiple expressions of interest, a diverse group of participants could be invited to the focus groups. These factors included participants’ gender, ethnic group, health and disability status, housing status, highest qualification and a question about whether the person supported vaccination in general. Participants were also asked to share their contact details with the researchers and give consent to be contacted. Further details about the study were shared with those eligible to take part in terms of age and geographical location, providing a copy of the participant information sheet and a link to an online consent form. We received 131 pre-screening questionnaires and 94 completed consent forms; 56 people took part in the FGs (see Table 2 below). Those who were eligible and consented but did not participate in the study mostly withdrew due to busy schedules or stopped responding to communications. All those who consented were invited to join a FG.

### 2.3. Data Collection

Participants who lived in the research locations were invited to take part in a FG. Participants were offered a range of times and dates in an effort to find mutually convenient times. We used Zoom video calling for the FGs and shared guidance for how to use Zoom with all participants. This included an explanation that Zoom can be accessed through tablets and smartphones, in addition to computers. Five of the FGs combined people across research locations due to a slow response rate. FGs lasted approximately 90 minutes and followed a topic guide (See Appendix A) in a flexible manner. Participants were offered a GBP 30 high street shopping e-voucher as compensation for their time. We felt that offering compensation for participation in FGs was unlikely to lead to bias because the group discussion setting meant one participant had limited influence over the overall discussion.

### 2.4. Data Analysis

Qualitative analysis was completed using QSR NVivo v12 software. Three transcripts were coded initially through line-by-line coding by TS before discussing the coding framework with the research team. The coding framework was then applied independently to two further transcripts by two researchers (CC and JK). Once the remaining transcripts were line-by-line coded, thematic analysis [26] was completed by TS and initial themes further discussed amongst the research team. The team noted that the themes captured phenomena occurring at the personal, interpersonal, community and system level (see Figure 1), and the socio-ecological model [27] was identified as a useful way to organise the themes.

## 3. Results

Data on participants’ views on vaccination and the rationale for these views were organised into four groups of themes considering attributes and interactions related to each aspect until thematic saturation was reached. For a list of subcategories, please see Appendix A.

### 3.1. Personal Factors Affecting Views on Vaccination

A number of personal factors were identified, which shaped the participants’ general views on vaccination and perceptions on the purpose of vaccines. These related especially to previous experiences of vaccination and healthcare and personal decision-making practices, which highlighted that participants believed that decisions about vaccination were primarily a personal matter.

#### 3.1.1. Decision Making

Most participants saw vaccination as an important way to protect themselves and others around them. This was the result of considering the potential risks and benefits through individual processes of decision making: “Any vaccinations I was just given them all the way through and I’m now a believer, what’s the evidence, what’s the upside, what’s the downside, read the sensible info and make your own decisions” (G9, P2, age 69).

The perceived benefits related to the need, as well as responsibility, to protect oneself and others. These aspects were linked to the potential to eradicate infectious disease and the importance of preventing the spread of infectious disease, which was emphasised in relation to family and friends through considerations of underlying conditions: “it [vaccination] does have an important role in protecting other people, and it can have a role in eliminating a disease. I think the same is true of polio, hardly anyone gets polio now, whereas pre-war, my mother’s generation…her stepbrother had it and he died young because of it” (G3, P4, age 71).

The participants generally considered the perceived benefits to outweigh potential negative aspects or discomfort related to vaccination. However, for the influenza vaccine, the perceived benefit, and therefore importance, was questioned because participants felt they were fit and well enough without vaccination. “I have chosen not to take up the influenza vaccines because I don’t suffer seriously from flu when it comes around so my own system seems to manage what happens in my environment and I encourage it to do that, rather than accept the offer of a vaccination” (G11, P3, age 69).

When discussing side effects, most participants referred to common or mild side effects, such as having a sore arm or feeling tired for a day or two. These, together with discomfort related to injections or needles, were acceptable for most, considering the expected benefits of getting vaccinated. (See Appendix A, quote 1).

Overall, many participants showed unquestioning support of vaccination in general, which suggests that getting vaccinated does not always involve internal deliberation or debate. Those participants who stated they prefer not to get certain vaccines provided more detailed accounts of why they felt some vaccines were not for them due to reasons such as previous adverse reactions or perceived strong health, emphasising that they were making a personal choice. (See Appendix A, quote 2).

#### 3.1.2. Previous Experiences

When discussing previous experiences of vaccination, nearly all participants had some memories of school-time vaccination. These experiences acted as a starting point for considering vaccination as routine practice. “In school I think you were frogmarched down the long corridor with a nurse at the end of it [laughs] or the doctor I can’t remember which. I did get mumps and measles [vaccines], both and probably jaundice as well” (G11, P1, age 69).

Previous experiences also included vaccination due to international travel and employment-related vaccination requirements, both of which contributed to a view of vaccination as normal and routine (see Appendix A, quote 3).

### 3.2. Interpersonal Interactions and Views on Vaccination

#### 3.2.1. Friends and Family as Sources of Information

Friends and family members were an important source of information for the participants in terms of vaccine availability or experiences of infectious disease. “I don’t know very much [about the vaccine programme], I have to say. I was aware there was one [a vaccine] for pneumonia, mainly because I knew someone who had pneumonia and there was some discussion related to that” (G8, P2, age 62). This also applied to awareness of the severity of infectious disease, which led to potential increased interest in getting vaccinated: “Having had a couple of friends suffer very badly with shingles in the last year or so… ‘Just give it to me’ kind of thing!” (G5, P1, age 67).

Many participants agreed that they shared the view about the importance of vaccines with others in their social circles, which was a reason why vaccines did not generate much debate. (See Appendix A, quote 4).

#### 3.2.2. Influencing Others

Many participants stated that they encouraged others to get vaccinated, but also considered vaccination a topic where you sometimes have to agree to disagree: “I tended to have conversations with people about flu jabs most years at flu jab time because where I was working it was—we were trying to encourage people to have the flu jabs and there’s some, ‘yes, yes I’ve always had it, yes, yes’. There’s others that say ‘oh no I never have it, I’ll not get flu, I’ll be okay’ (G9, P3, age 59).” Despite the inability of some participants to understand vaccine hesitance in others, the participants were broadly in agreement that not getting vaccinated was a personal choice. “Yes—we debated quite a bit because there’s ‘For-s’ and ‘Against-s,’ and we have to respect one another” (G12, P1, age 68). In instances where the participants had encountered people with differing views, many emphasised that they were not prepared to sacrifice their social relationships due to disagreements about the importance of vaccines. (see Appendix A, quote 5).

### 3.3. Wider Community and Cultural Aspects

This group of themes focuses on the influence of people not directly known by the participants, hence, referring to the wider community and culture. The participants’ interactions within their wider communities reflected cultural and socio-demographic aspects, such as age, religion and ethnic background.

#### 3.3.1. Generational Differences

The participants believed that their vaccination behaviours were different to those of younger people. Participants tended to characterize younger generations by individualism, but especially a need to question general advice and seek further information, enabled by the availability of the Internet and social media. In contrast, participants saw their generation—and that of their elders—as those who tended to respect and trust public health advice, especially the NHS. The importance of vaccination due to a need to act in support of the greater good was emphasized as something the younger generations may be lacking: “We’re used to the National Health Service and we’re looking for more services from it rather than less… I’m just guessing here that my children’s generation are the Dr Google generation where they have maybe way too much information and they’re Googling their health whereas I’m coming from automatic pilot. I just accepted that people did things to me and they were trying to help me” (G10, P5, age 77). (See also Appendix A, quote 6).

#### 3.3.2. Cultural Aspects

Religion and ethnic background were rarely discussed by the (mainly White British) participants, despite being asked about beliefs or habits that might affect their vaccination decisions (see Appendix A). Ethnicity was mostly mentioned in relation to media coverage and the participants’ concerns about lower vaccine uptake among certain ethnic groups, including their acquaintances (see Appendix A, quote 7).

Some participants expressed concerns over friends or acquaintances who had shared with them that they preferred not to get vaccinated due to their religious beliefs. A small number of participants stated they were religious themselves or held an active role in their local church; however, they stated this did not discourage them from getting vaccinated, but emphasised instead the importance to protect their community. One participant in Northern Ireland felt that whether they were vaccinated as a child or not was potentially linked to their mother’s religious beliefs (See Appendix A, quote 8).

Those participants who were generally supportive of vaccination held strong views about those who prefer not to get vaccinated, especially regarding the COVID-19 vaccines. However, some participants also emphasised that each individual was within their rights to decline a vaccine should they choose to do so. “I’m listening to the annoyance in people’s tone around people who won’t have the vaccine. I’m in an interesting situation with it myself, although I’ve had the vaccine, I love a lot of people who won’t have the vaccine… They have a right to [decide] what goes into their body, especially when something is at research stage, so I stand by them” (G4, P2, age 55). (See also Appendix A, quote 9).

#### 3.3.3. Alternatives to Vaccination

Regardless of whether the participants were supportive of vaccination or not, all agreed that there were precautions each individual could follow to reduce the likelihood of illness or serious aftereffects related to infectious disease. Many of these related to recommendations made during the COVID-19 pandemic: the use of facemasks, regular hand washing, avoiding touching one’s face, cleaning surfaces and adequate ventilation indoors. Other aspects related to diet and lifestyle: regular exercise, nutritious food and vitamin D supplements. “I believe the vitamin D is now acknowledged as being useful in boosting the immune system… I think that keeping up your general health is the most important thing that you can do other than get the vaccination if that is your wish” (G13, P3, age 79). More restrictive precautions related to limiting social contacts, social distancing in public spaces and isolating during illness (see Appendix A, quote 10).

### 3.4. System-Level Aspects

#### 3.4.1. Access

Accessing vaccines was easy for most participants; we did not find differences in terms of access to vaccines in urban or less urban areas. The exceptions to access included obtaining certain travel vaccines, a brief disruption to influenza vaccine provision or an unusual need to travel long distances to access one of the COVID-19 vaccines. There were some individual preferences in terms of where the participants preferred to get vaccinated. Some felt that their general practice was the most appropriate place to get vaccinated, and for many, it was what they were used to. “I was offered the shingles jab about three years ago when I got my flu jab, I got them at the same time, and then the next year when I went for my flu jab they gave me the pneumonia one at the same time… It’s almost a non-event… I haven’t had to ask for them, they just said ‘do you want one?’ and I said ‘yes, I’ll take it’” (G4, P3, age 76). Some considered pharmacies a more convenient way to get vaccinated due to shorter waiting times and central locations. “It would seem to me pharmacists are taking on a greater role in all sorts of healthcare, certainly in Scotland… And that would seem to be a convenient way for people” (G8, P2, age 62). Discussions about pharmacies as a venue for vaccination led to some debate about the privatization of the NHS, suggesting that the convenience they offered perhaps had a broader social price tag attached to it.

Access barriers related to GP access more broadly, as many stated it was difficult to get a GP appointment, let alone find opportunities to discuss vaccines with an HCP. “You never get to see your GP… The GPs they haven’t got time, and you can’t get through to them anyway, so, you’re not discussing with them” (G3, P6, age 89). This was not the case in some of the less urban research locations, such as Banbridge in Armagh City, Banbridge & Craigavon; Perth & Kinross or Bedford; or Newham in London.

#### 3.4.2. Lack of Information and Sources of Information

The data show lack of information, as well as unreliable information and misinformation, as the key challenges to vaccine uptake. Lack of information related to an inadequate provision of information or vaccine offers from healthcare providers regarding the vaccines available to the participants. “Can I say—I’m 83, and I haven’t been offered either of those [shingles and pneumococcal vaccine]. There was a notice in the surgery saying about shingles, and I said, ‘Does that apply to me?’ ‘No—you’re too old.’ They told me I was too old at 83, so why? As for pneumococcal—no offer at all” (G1, P1, age 83).

Where the participants were offered vaccines by healthcare providers, this was often done in an opportunistic manner during a visit for a different matter. While some felt this was convenient, some were discontented with the level of information they were provided about vaccines and the lack of opportunity to discuss their relevance. “I suppose, if somebody said, ‘Well this is…’ You only need it [pneumococcal vaccine] once, don’t you? You don’t have to have it annually—you don’t have to have it boosted—and so, ‘This is a one-off, and it’ll last you as long as you like’—that’s probably a good selling point that was missed at the time for me. You know, save you coming back, and it’s not necessarily a repeat one” (G1, P2, age 70). The participants were also unsure about how a disease such as shingles might affect them, which can affect whether they think the shingles vaccine is relevant to them.

We also found consistent differences in whether or not the participants had been informed about the vaccines available to them by their HCPs or if they had been invited to get vaccinated. Many participants expressed their concerns about differences in vaccine offers: “I’ve got a friend who’s a bit older than me, we live in the same village but we’re at different practices. I got my pneumonia [vaccine] when I was 65, and she’s never been offered it, so it just seems strange that we’re both in the same area… yet we’ve got two different GPs with different criteria” (G7, P4, age 66). However, this did not apply to the COVID-19 vaccines, which one participant described had been rolled out as a “well-oiled machine” (G1, P4, age 58).

#### 3.4.3. Policy

Participants expressed some discontent with the overall public health messaging around the vaccination programmes available to older adults. This related to the perceived inconsistencies in vaccinations offered through HCPs. Some participants questioned why the shingles vaccine was offered only at 70 years and not earlier. When asked if they felt it was appropriate to vaccinate older adults, many felt that limiting the current free vaccine offer would not be well received. “Well, presumably, there are still medical reasons why we *should* be actively as a group having these vaccines, and it seems to me… the flu is plastered everywhere imaginable, COVID etc. Yet, shingles and pneumonia seem to be, not just the hidden diseases, but the vaccines are hidden as well” (G3, P5, age 65). (See also Appendix A, quote 11).

The participants shared concerns about the state of the NHS and felt vaccination could prevent overburdening the public healthcare system further. While there were no clear regional differences in the data, the participants in Northern Ireland and Scotland brought up the political side of public health messaging, which relates to who is considered to hold the highest authority in the devolved nations when it comes to urgent health messaging.

## 4. Discussion

This research aimed to increase the understanding of older adults’ perceptions of the influenza, shingles, pneumococcal and COVID-19 vaccines that currently form the government-funded vaccine offer for older adults in the UK. The findings show that the older adults we spoke to were generally supportive of the vaccines available to them and saw vaccination as a way to protect themselves and others around them. Their views on vaccines relied heavily on their previous experiences and information from their interpersonal interactions, whereas HCPs’ role in supporting vaccine decision making was limited. While the people we spoke to were broadly supportive of vaccination, either awareness of a vaccine’s availability and how to request it or an offer of a vaccine by HCPs is required in order to receive it. This is particularly true for shingles and pneumococcal vaccines, which are less well publicised than the influenza vaccine offer. 

The limitations of this study include the limited diversity of the research participants in terms of ethnicity; the limited number of participants who were against or less supportive of vaccination; and the small number of participants in some of the FGs, which limited the total number of participants. We were unable to recruit many participants from minority ethnic backgrounds, despite efforts to recruit in ethnically diverse areas and attempts to liaise with community groups working with different ethnic groups. Using recruitment materials that were aimed at those who preferred not to get vaccinated only resulted in one further FG. For these reasons, our findings are not reflective of less vaccine-confident people and do not represent the views of minority ethnic groups in the UK. There were two instances when participants did not attend a scheduled FG and one last minute withdrawal, which resulted in three FGs consisting only of three participants. However, a small number of participants per group was also beneficial because the small group setting enabled increased involvement from all who attended. We aimed to hold 15 FGs, with up to 7 participants in each FG, but were unable to recruit this many participants, despite recruitment being active for 11 months.

The strengths of this study lie in the qualitative approach to research, with people aged 55+ as a broader age group across the whole of UK, in addition to focusing on a full public health vaccine programme instead of a limited age range or a specific vaccine. This enables an in-depth understanding across a vaccination programme as a whole for this group, providing information that can support enhancing current vaccination delivery. As far as the authors are aware, such a study has not been completed before.

Participants’ decisions to get vaccinated or not appeared to be mostly based on personal decision-making processes. These processes appear to be affected by previous experiences with vaccines and diseases, including childhood vaccinations. Previous experiences appeared to contribute to forming positive views on vaccines, as well as to perceiving vaccination as a routine procedure. Similarly, previous studies show that previous positive experiences relate to vaccination intentions [8,18,21]. However, few studies discuss childhood experiences and the development of a view of vaccines as normal and routine, even if experiences and values are acknowledged as a factor [18]. It should also be noted that some older adults who were born outside of the UK and migrated to the country at a later stage may have different experiences of childhood vaccinations [28].

Interpersonal interactions appeared to support individuals’ decision-making processes, with a sense that opportunities to talk about vaccines could lead to people expressing intentions to get vaccinated. Receiving vaccine information has been shown to lead to positive vaccine intentions [2], whereas interactions with lay people have been linked to inaccurate interpretations of infectious disease [18]. This emphasises the importance of accessing impartial and accurate information about vaccines and infectious disease to support informed decision making. Our findings show that difficulties accessing an HCP, as well as a lack of information, can limit vaccine uptake. Previous research shows that recommendations from HCPs [3,8], frequent visits (3+ per annum) to a general practitioner [9] and being presented with “non-biased information” can improve vaccine uptake [2], while positive experiences of primary care can improve the acceptance of vaccines [6]. While previous research has focused on the benefits of vaccination provision through pharmacies [29,30], our findings highlight specific individual preferences in terms of where older adults prefer to get vaccinated, with some preferring their general practice, while others find pharmacies more convenient.

Overall, the data suggest that HCPs’ role regarding vaccine information provision should be expanded if vaccine uptake is to be improved. Furthermore, it should be explored at what point before a vaccination offer is made should information be shared and which means—such as leaflets, text messages or letters—would be appropriate, as well as cost-effective ways of sharing general information about vaccines. This should include understanding the inconsistencies in why some people seem to be offered vaccines while others are not, despite being the same age and registered at the same general practice.

Further research is required to explore whether there is variation in how vaccines are offered in the UK and how this could be addressed to support improving vaccine uptake. This study has shown especially how people aged 55+ form vaccine-supportive views. However, it should be explored whether these patterns may inform behaviours among people who prefer not to be vaccinated. Whilst the aspect of mistrust has been discussed in relation to vaccine hesitancy [20], less is known about how to build trust in HCPs or the broader healthcare system and whether that could improve vaccination uptake among vaccine-hesitant individuals.

Recommendations from this research include improving the provision of information and encouraging opportunities to discuss vaccines and infectious disease in healthcare settings to ensure that older adults are able to make better informed decisions regarding the vaccines available to them. A national effort to improve especially the provision of information about the pneumococcal vaccine and shingles vaccine would be beneficial, along with information about shingles as a disease. Furthermore, general information about vaccines should be made available before any vaccine offer is made by HCPs to reduce the need for impulsive decisions. Increasing the provision of vaccines through community pharmacies should also be carefully considered because of the differing preferences individuals have when accessing vaccines.

The current vaccine offer is based on an understanding of the spread of infections and expected benefits of disease prevention, but little is known about the reasons why uptake varies among older adults as a target population. The findings of this research show that opportunities for informing and discussing the vaccines that are available to older adults is valued, as this can help individuals make better informed decisions about vaccines, regardless of whether they choose to get vaccinated or not.

## Figures and Tables

**Figure 1 vaccines-11-00870-f001:**
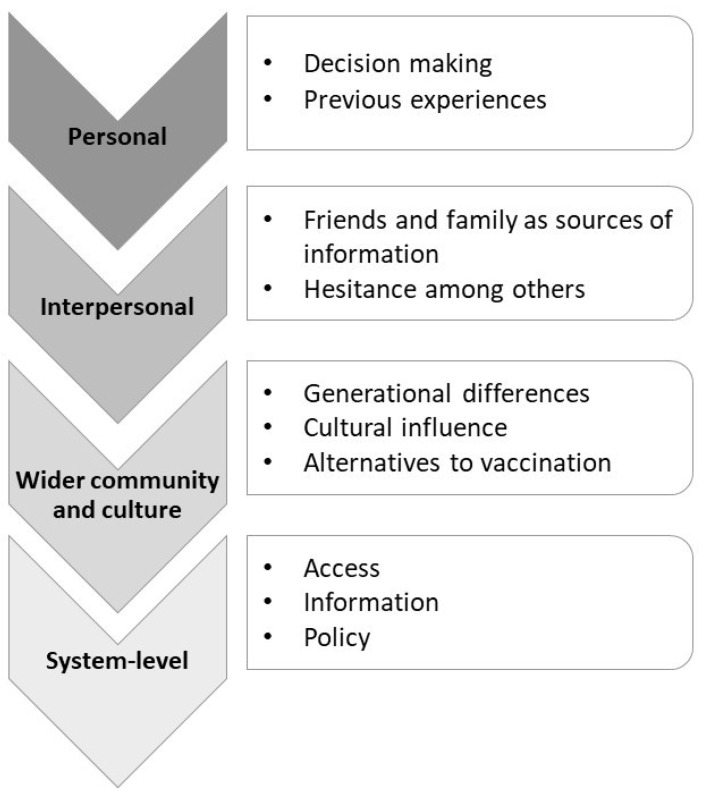
Four main groups of themes.

**Table 1 vaccines-11-00870-t001:** List of research locations by country. Urban areas are marked with *.

Local Authority	Country
Bedford	England
Birmingham *	England
Bradford *	England
Manchester *	England
Newcastle-Upon-Tyne *	England
Newham borough of London *	England
Tewkesbury	England
Wellingborough	England
Wycombe	England
Armagh City, Banbridge and Craigavon	Northern Ireland
Belfast *	Northern Ireland
Glasgow *	Scotland
Perth & Kinross	Scotland
Cardiff *	Wales
Pembrokeshire	Wales

**Table 2 vaccines-11-00870-t002:** Participant characteristics.

	Number (n)
**Age group**	
55–64	19
65–74	29
75–84	6
85+	2
**Gender**	
Female	33
Male	22
Non-binary	1
**Ethnic group**	
White	52
Mixed ethnicity	2
Asian	1
Black	1
**Health status**	
In good general health	28
No serious underlying health condition	18
Underlying health condition	1
Serious underlying health condition	7
Other	2
**Disability status**	
Disability	10
No disability	43
Prefer not to say	3

## Data Availability

The data presented in this study are available on request from the University of Bristol data repository, data.bris.ac.uk. The data are not publicly available due to ethical restrictions.

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
