# Peer review of "Views and Perceptions of People Aged 55+ on the Vaccination Programme for Older Adults in the UK: A Qualitative Study"

_vaccines, 2023, doi:10.3390/vaccines11040870_

Round 1

Reviewer 1 Report

This manuscript deals with "Views and perceptions of people aged 55+ on the vaccination programme for older adults in the UK: A qualitative study".  Study is very interesting and i suggest a minor correction and require a detailed clarification. Correction to be addressed by the authors as follows: The abstract is not well organized, where the sentences are incomplete and no continuity is there. It would be feasible, if include the significance of the current study in the abstract. A brief description of how the authors selected information from the literature in the databases, as well as what time period they searched for, is missing. Authors should justify and expand the information on the advantages of this study. Authors should specify the main experimental conditions used on the evidences from the literature. Where they briefly describe the most important data reported in the literature in a homogeneous manner and sequence reinforcing the relevance of of this approach. Authors should discuss whether the use of this method represents a solid alternative to existing methods.

Conclusions should reaffirm the fundamental contribution of this paper.

Reviewer 2 Report

Silvonen et al studied the perspectives of COVID-19 vaccine response in older adults. The authors took good efforts to organize older people and took their views. However, it seems the study has little bias towards only vaccine-positive views are represented without the people those do not support vaccination.  Before publications, these are the following concerns that should be explained more distinctly in the text.

1.       Study group is too small (n=56). Higher number would have been recruited.

2.       As the study groups are old age people >60yrs), most of them probably nonworking and paid £30, author should include/discuss nonbiasness for their answers.

3.       The study group mostly consists of white people (52/56), how the results  would vary from other ethnic groups? When in UK, mostly other ethnic groups than white population are mostly affected by COVID-19 related death and how their perception about intaking vaccines.

4.       The most weakness of the study is stated by authors themselves in line 369-372. These are the major factors influencing the study.

5.       The results of positive outlook of vaccination could be biased as the study groups are mostly vaccinated.   Views from nonvaccinated groups are mostly missing.

6.       Also, the positive outlook of vaccination mostly came from the individuals with pre-vaccinated people in their childhood. But in UK, lack of information from other ethnic groups, mostly migrants with no vaccination history.   
